# Self-management of depression and anxiety amongst frail older adults in the United Kingdom: A qualitative study

**Pushpa Nair** [1]*, **Kate Walters**[1], **Su Aw**[2], **Rebecca Gould**[3], **Kalpa Kharicha**[4], **Marta College Buszewicz**[1], **Rachael Frost**[1]

**1** Department of Primary Care and Population Health, University College London Medical School (Royal Free Campus), London, United Kingdom, **2** National University of Singapore, Saw Swee Hock of Public Health, Singapore, Singapore, **3** Division of Psychiatry, University College London, London, United Kingdom, **4** NIHR Health & Social care Workforce Research Unit, The Policy Institute, King's College London, London, United Kingdom

* pushpa.nair@nhs.net

**Data Availability Statement:** This is a qualitative study and therefore the data generated is not suitable for sharing publicly beyond that contained within the report due to potentially compromising

## Abstract

### Objectives

Depression and anxiety are common in frail older people and are associated with high levels of morbidity and mortality, yet they typically face greater barriers to accessing mental health treatments than younger people and express preferences for self-managing their symptoms. This study aims to explore frail older adults' experiences of self-managing symptoms of depression and/or anxiety.

### Design

Qualitative semi-structured interviews, exploring experiences of depression and/or anxiety, ways participants self-managed these and the contexts within which this took place. Interviews were audio-recorded and transcribed verbatim.

### Participants

28 frail older adults in the United Kingdom, purposively sampled for neighbourhood, frailty and symptoms of anxiety/depression.

### Analysis

Thematic analysis to inductively derive themes from the data.

### Results

Our findings suggest that frail older adults find maintaining independence, engaging in meaningful activities, and socialising and peer support important for self-managing depression and anxiety. These could all be adapted to the level of frailty experienced. Drawing on life experiences, addressing the perceived cause and faith were helpful in some situations and for some personalities. Distraction and avoidance were helpful for more severe

anonymity by revealing identifying or sensitive participant information. Further information can be obtained from the corresponding author. Details for the ethics committee for this project are: NHS Camden and Kings Cross Research Ethics Committee (ref 17/LO/1963), camdenandkingscross.rec@hra.nhs.uk.

**Funding:** The National Institute for Health Research (NIHR) School for Primary Care Research sponsored this research [https://www.spcr.nihr.ac. uk/]. Grant number 378 (awarded to PN, KW, RF, RG, KK, MB, RF). The funder had no role in study design, data collection and analysis, decision to publish, or preparation of the manuscript.

**Competing interests:** The authors have declared that no competing interests exist.

symptoms or where the causes of symptoms could not be resolved. Self-management strategies were less well-established for anxiety symptoms, especially when linked to newer health fears and worries about the future.

## Conclusions

Developing services and sources of information that support and facilitate key therapeutic components of self-management, which align with older adults' preferred coping styles and take into account levels of frailty, may be a way of supporting frail older people waiting for mental health treatments or those who prefer not to access these. Greater awareness of anxiety and how it can be self-managed in frail older people is needed.

## Introduction

Worldwide, approximately 11% of adults aged 65+ years are frail [1]. Although lacking a universal definition, frailty is generally conceptualized as a dynamic process involving deficiencies across multiple physiological systems, with diminishing reserves and increased vulnerability to poor health outcomes [2], encompassing symptoms such as weight loss, muscle weakness, low physical activity and low energy levels [3]. However, psychological aspects of frailty are often overlooked [4].

Depression has a prevalence of 9% in adults aged 75+ years [5], and anxiety 15% in those aged 60+ [6]. Frail older adults are four times more likely to experience anxiety and/or depression than non-frail older adults [7]. Both are associated with higher levels of functional impairment [8, 9], cognitive decline [9–11] and healthcare service utilization [12, 13], especially when combined with frailty [14], and both remain commonly under-diagnosed [15]. Depression in older adults has additionally been linked to greater risk of falls [16] and self-neglect [17], with a reciprocal relationship with frailty [18] and higher relapse rates than in earlier life [19]. The mental health of older adults has become an even more urgent concern considering the negative impact of the Covid-19 pandemic on wellbeing in this group [20].

Uptake of mental health services amongst older adults is poor—up to 70% of adults aged 55 + in the United States (US) with mood disorders do not access mental health services [21]. Within the National Health Service (NHS) in the United Kingdom (UK), mental health services can be freely accessed through a GP referral or self-referral; however only 6.2% of national referrals to psychological therapy services within UK primary care settings are for those aged 65+, and this drops to 2.1% for those aged >75 [22], with an even greater reduction in referrals seen during the recent Covid-19 pandemic [23]. There is evidence that many older adults view depression as a normal part of ageing and often prefer self-management approaches over seeking professional help [24, 25]. Practical difficulties such as limitations in mobility and transport costs have also been cited as contributing to lower levels of access, particularly amongst frailer older adults [26]. There is evidence that frail older people prefer to access mental health treatments through their GP, but experience problems accessing their GP or finding information from alternative sources [27]. In addition, there is evidence that healthcare professionals experience a wide range of barriers in referring older people to psychological therapies [27]. Those who are referred for psychological therapies can find they are on waiting lists as long as 12 months in some areas, with little support available to them whilst waiting for this service [28]. Waiting times have been further exacerbated by the Covid-19 pandemic [23], with certain services only offering remote access, potentially leading to the digital exclusion of some older adults [29].

The value of current treatments for mood disorders in frailer populations is also uncertain. The efficacy of antidepressants is less clear in frailer populations [30], with multiple medications a known contributor to frailty itself [31]. Despite this, there is evidence that the oldest old are more likely to be prescribed medication than be referred for psychological therapies compared to those who are younger [32]. Psychological therapies in frail older people may also be less effective, may need to be adapted to account for mild cognitive impairment [33, 34], or may be less acceptable; for example, a US survey showed that those aged 85+ preferred supportive treatment for anxiety rather than Cognitive Behavioral Therapy [35].

In this context, self-management strategies may play a significant role in improving mood disorders in this population [36]. Self-management is defined as 'taking increased responsibility for one's own health, behaviour and well-being' [37]. It can encompass lifestyle changes, problem solving, goal-setting and symptom monitoring [38, 39]. It is important to distinguish self-management, which focuses more on maintaining wellbeing and reducing relapses [39], from self-help, which aims to treat episodes of depression or anxiety using remote or instructional CBT-based approaches [40]. Whilst help-seeking is also a part of self-management, barriers to this have been extensively documented in older people [25, 41]. Ways that older people can support themselves align with a preference for feeling independent at a time of increasing dependence [27], fear of stigma and a potential inclination to deal with emotional problems on their own [25, 42, 43]. Self-management thus has the potential to improve older adults' quality of life by facilitating feelings of empowerment and self-efficacy [38, 42].

Self-management in older populations has most commonly been studied in connection with long-term medical conditions [43–45], with studies showing good self-management outcomes. Programmes adapting the collaborative chronic care model (CCM), which includes self-management as a key component, to mental health care [46] have also shown promising outcomes for depression across all age groups [47–49]. However, most studies specifically exploring self-management of mood disorders have focused on younger populations and found that a wide range of strategies are used [50, 51], such as physical activity, social engagement, being proactive and having knowledge about the condition [39, 52].

Limited qualitative literature exploring aspects of self-management of mood disorders in older adults have focused mainly on depression in 'younger old' (<75 years) non-frail groups, highlighting the importance of strategies such as socializing, leaving the house (e.g. to go shopping), engaging in religion, exercising and keeping up hobbies (findings summarized in three qualitative meta-analyses [25, 53, 54]). More recent studies [41, 42, 55, 56] have highlighted that a sense of identity, meaning and experiential knowledge are particularly important for older adults' capacity to self-manage depression. There is much less literature exploring self-management techniques for *anxiety* in older adults. Furthermore, we know little about the capacity and motivation of *frail* older people to self-manage depressive symptoms, despite the higher prevalence of symptoms in this subgroup, and much less about how they view and self-manage anxiety symptoms. Knowing more about their self-management techniques may help develop supportive resources that recommend acceptable strategies for alleviating mood symptoms in this population.

In this study, we therefore aimed to explore self-management strategies for both anxiety and depression amongst the frailer 'older old' group (75+ years), for whom some strategies (e.g. exercise, socializing) may be compromised by reduced mobility and multiple health comorbidities.

## Research design and methods

The voices of frail older adults are often absent in research [57]. We therefore chose a qualitative methodology that sought to centre the experiences of frail older adults, to better

understand facilitators and barriers to self-management strategies, how frailty influenced these and how strategies used compared to those of younger-old adults.

## Ethics statement

The study was approved by NHS Camden and Kings Cross Research Ethics Committee (ref 17/LO/1963) and research was carried out in accordance with the ethical principles for medical research involving human subjects expressed in the World Medical Association Declaration of Helsinki [58]. Informed written consent was obtained for all participants involved in the study.

## Participants

We recruited 28 participants in total from five UK general practices, across one semi-rural (n = 10), one suburban (n = 4) and three urban areas (n = 14), in order to sample neighbourhoods with different ethnicities, socioeconomic backgrounds, and variable access to mental health services. The UK has a free national healthcare system (NHS) and referrals to mental health services are primarily accessed through general practice, although patients are also able to self-refer. Data was collected in 2018.

## Inclusion criteria

Practice staff and general practitioners (GPs) searched their patient lists to identify older adults (aged 75+), classed as moderately to severely frail according to their practice's frailty register, who were currently experiencing symptoms of depression and/or anxiety. We asked practices to include a mix of those with and without a previously recorded diagnosis of depression and/ or anxiety in their medical notes, as depression and anxiety are often under-diagnosed in this population and we wanted to also include those who were currently experiencing symptoms but did not have a formal diagnosis and had not accessed previous help.

## Exclusion criteria

Older adults with advanced dementia, and those who had <6 months life expectancy were excluded.

## Procedure

Practices posted invitation letters (with pre-paid postal reply slips) to patients who met the above inclusion criteria, and included study information leaflets containing brief screening questions for symptoms of anxiety and depression to encourage participants to self-identify (adapted from the 2-item Patient Health Questionnaire and 2-item Generalized Anxiety Disorder scale, which are both validated in older populations [59, 60], but shortened and with the 2-week timescale changed to 'recently'). Interested participants returned reply slips to the two primary investigators (PN and RF) and were then telephone-screened to ensure they met the inclusion criteria, by using the same questions as in the leaflet to confirm symptoms, and asking about functional difficulties to confirm frailty. Of the 263 postal invites sent out, we received 41 positive responses, of which 28 were deemed eligible for study inclusion through screening, and 42 negative responses.

## Data collection

We collected data through individual semi-structured interviews, conducted face-to-face at participants' homes (or another location if requested). Interviews were conducted by PN

(n = 9, an academic GP, not disclosed to participants unless directly asked) and RF (n = 19, a health services researcher). Terms such as 'low mood' and 'worry' were used at the beginning of interviews to reduce any perceived stigma, but terminology was adapted to reflect participants' own language as interviews progressed. Each interviewer observed 1–2 of the other's interviews to ensure a consistent approach. Interviews lasted approximately one hour (range 34–89 minutes) and were audio-recorded. The interview topic guide was developed with the input of the whole research team, including our two patient and public involvement (PPI) representatives (who were older adults from different ethnic minority backgrounds interested in mental health research in older people) and was modified as interviews progressed. We asked interviewees about their typical week, past and present experiences of low mood and anxiety, any coping and self-management strategies, social/carer networks, and their views on treatments, help-seeking and suggestions for improvements in mental health services (the full topic guide can be found in [61]). As there were rich data across a wide range of themes during interviews and subsequent coding and analysis, this current paper focuses on the contexts and forms of self-management of mood symptoms. Data related to help-seeking, mental health care treatments and service improvement have been published elsewhere [61]. At the end of each interview, we collected demographic data. Participants were provided with a list of relevant support services, an Independent Age depression guide [62] and a £20 voucher. Audio-recorded interview data were transcribed verbatim by an external transcription service, and anonymized and verified for accuracy by the interviewer.

## Data analysis

We used thematic analysis to analyze our data from a constructivist perspective, which emphasizes the importance of multiple perspectives, contexts and values [63]. Our team included two health services researchers (RF, KK), three academic GPs (KW, MB, PN), two psychologists (RG, SA) and two PPI members (WD and SL). All transcripts were read by PN and RF and at least one other team member. PN and RF developed a thematic framework (which can be found in [61]), which was refined at a team meeting. PN, RF and SA coded transcripts line-by-line using NVivo 12 [64], according to the thematic framework. After coding three pilot transcripts, PN, RF and SA met to check for coding consistency and refine the thematic framework. Relevant headings from the thematic framework included: current experiences of depression/anxiety, influencing contextual factors, impact of physical health/frailty, life history and self-management strategies. PN and RF had regular meetings with the team to discuss, agree on and refine emerging analytical themes. Saturation was considered to have occurred after 26 interviews, with minor and wider collective themes accounted for, and two further interviews were conducted to corroborate this.

## Results

We recruited 28 participants with a mean age of 80.71 (SD 4.07, range 75–88). Participants were approximately two-thirds female (n = 19) and predominantly White British (n = 22). Ten participants were receiving Pension Credit, an indicator of low income in the UK (see Table 1 for demographic details). Whilst only eight participants reported receiving a formal diagnosis from a clinician (in their medical notes) of depression, anxiety or both, telephone screening identified nearly half the participants as experiencing current symptoms of both anxiety and depression, with the rest experiencing one or the other.

Participants reported a range of experiences of depression, ranging from transient periods of feeling down, which varied depending on physical health and levels of functioning, to constant levels of low mood, which were often underpinned by family conflict, financial

**Table 1. Demographic characteristics of study participants.**

| Demographics | Number of Participants | |
|---|---|---|
| Age mean; range | 80.71 years; 75–88 years | |
| Gender (male:female) | 9:19 | |
| Neighbourhood | Urban | 14 |
| | Suburban | 4 |
| | Semi-rural | 10 |
| Ethnicity | White British | 22 |
| | Black Caribbean | 4 |
| | Indian | 1 |
| | White Irish | 1 |
| Country of birth | United Kingdom | 22 |
| | Ireland | 1 |
| | Trinidad | 2 |
| | Jamaica | 2 |
| | Mauritius | 1 |
| Ethnicity | White British | 22 |
| | Black Caribbean | 4 |
| | Indian | 1 |
| | White Irish | 1 |
| Marital status | Married | 8 |
| | Widowed | 10 |
| | Separated | 2 |
| | Divorced | 7 |
| | Single | 1 |
| Living situation | Alone | 17 |
| | With spouse | 8 |
| | With other family | 3 |
| Housing | Owner-occupied | 16 |
| | Council rented | 7 |
| | Housing association rented | 2 |
| | Sheltered housing | 2 |
| | Private rented | 1 |
| Age completed education | Before 15 years | 8 |
| | 15–17 years | 7 |
| | 17–20 years | 6 |
| | Over 21 years | 7 |
| Pension type | State | 28 |
| | Employer | 17 |
| | Private | 3 |
| | Pension credits | 10 |
| Diagnosis (self-reported) | Depression | 4 |
| | Anxiety | 4 |
| | Depression & Anxiety | 3 |
| | No diagnosis | 16 |
| | Unsure | 1 |
| Telephone screening | Depression symptoms | 10 |
| | Anxiety symptoms | 5 |
| | Both | 13 |

difficulties or past traumatic life experiences. In terms of anxiety, some participants reported frequently worrying about 'silly little things', whilst others had strong health-related fears or fears about the future, and a few reported constant levels of worry about everything. As many participants experienced both anxiety and depression, it was sometimes difficult to distinguish differences in the way that they self-managed each, but this is highlighted where possible in the results.

Within the contexts of their frailty, participants described a wide range of strategies that helped them cope with symptoms of low mood and/or anxiety. They had developed these over their lifetimes and were able to implement them according to their current levels of functioning. This paper reports on the wide range of approaches used by participants to self-manage symptoms of depression and anxiety, incorporating four key themes: 1.) Maintaining independence; 2.) Meaning-making and recreation; 3.) Socializing and peer support; 4.) Internal mental and emotional coping strategies.

## Maintaining independence

Participants reported changes in perceived identity resulting from functional decline, illness and fears over losing independence, which negatively affected their mood and wellbeing:

> '*Old age isn't like 'flu. You get worse, like. That's what worries me, and I think I don't want to be a nuisance.*' (White British Female, age 81–85, depression and anxiety symptoms)

The ability to maintain some level of independence, despite experiencing frailty, emerged as integral to many participants' sense of identity. Even if participants could not do some of the activities they previously enjoyed, such as reading or exercising (due to deterioration in eyesight or limitations in mobility), being able to socialize, maintain a routine or keep the house tidy were all seen as representative of their ability to continue with 'normal' life.

> '*The more you go out, the more you become a normal, still remain a normal, active person*'. (White British Female, age 81–85, anxiety symptoms)

Despite these activities often being more difficult to uphold or having to be adapted as people became frailer, they provided some structure to the day and were used by some to combat low motivation and keep negative thoughts at bay.

> '*I've got a routine that I keep each morning. I do that, not because I'm pedantic, but because if I don't, I'm letting everything slide, you know.*' (White British Male, age 86–90, depression symptoms)

Family, neighbours and carers were generally identified as helpful in assisting them to keep to a routine and therefore supporting mental wellbeing. However, this needed to be carefully balanced with not appearing too dependent, and some feared that sharing any hint of being unable to keep up these activities with their families would either worry them or result in loss of their independence.

> '*I want to do things for myself, continue as long as I can, you know. I don't want them (family) running and worrying and whatever. . .*' (Black Caribbean Female, age 76–80, anxiety and depression symptoms)

**Meaning-making and recreation.** In addition to maintaining a sense of independence, activities that promoted a sense of meaning were also important for mood and wellbeing, particularly for participants whose frailty limited their ability to go out. Many mentioned not being able to use strategies that were previously helpful in managing their mood due to changes in functional levels or occasionally because of caring responsibilities. Creative activities, such as poetry, painting, writing, patchwork, music and photography, helped participants express and make sense of their emotions, and often helped to contextualize earlier life experiences and memories.

*'That's why I think I like patchwork because you use old material, you know, and there's a lot of memories in the old material'.* (White British Female, age 81–85, anxiety symptoms)

*'I also write dotty poems and things like that, I've got quite a collection of them. . .But when I feel, you know, in that mood, sometimes it comes out that way.'* (White British Male, age 86–90, depression symptoms)

Some male participants in particular, had taken on new self-directed recreational projects, which were often sedentary and sometimes facilitated by the use of technology; for example, researching local history, programming or constructing family trees.

*'Everyday, most days, and I'm doing the family history. I'm just tracing right back to my ancestors.'* (White British Male, age 86–90, depression symptoms)

Searching for meaning could manifest itself as a need to be productive, have a sense of purpose or contribute towards something, for example by volunteering or finding a new project in life.

*'Maybe if some people, instead of sitting down and twiddle their thumb, could find something to do. You know, it would help. An interest. Find an interest in life.'* (Black Caribbean Female, age 86–90, depression symptoms)

*'And I think they should do more things, not silly things, not like birthday cards and like that, but things–even if you're knitting squares for people all to sew together, to send to Africa or something, you know.'* (White British Female, age 81–85, anxiety symptoms)

The desire to be productive could offset feelings of dependency, associated with participants' frailty and low mood. However, these wishes were sometimes expressed as hypothetical statements rather than as practical plans, either because participants were not aware of available services or organizations that could help them realize their aspirations or because their frailty prevented them from doing so.

*'I mean if I could help somebody else you know, I feel, you know, and then probably that would help me'.* (White British Female, age 76–80, depression symptoms)

*'I mean I do want to get out of this, you know, and sort of to feel. . .like a normal person sort of thing, doing things or helping somebody. That's the thing. Instead of just somebody helping me you know.'* (White British Female, age 76–80, depression and anxiety symptoms)

## Socializing and peer support

**Camaraderie and connectivity.** Despite participants' frailty, many found ways to stay connected to family and friends, for example via the telephone or through attending local

groups, while a few also used social media, particularly if they had difficulties going out. Many participants cited speaking to friends, peers and spouses as useful forms of informal support. The value of this seemed to be underpinned by a sense of shared life experience and camaraderie, both in reviewing the past but also in facing the existential uncertainty of the future:

*'But I think most of us are very philosophical about things like that, because we're together and we're in the same boat. So we laugh about it, you know. Sometimes the humor is very black. But, it helps, you know.'* (White British Male, age 86–90, depression symptoms)

Many valued the reciprocal exchange of advice, as it drew on personal experiences and facilitated a sense of shared experiences. In addition to supporting mental wellbeing, exchange of advice around services providing help with practical things, such as finances, could also help people to overcome problems negatively affecting their mental health.

*'So a lot of connection happens out of just, you know, pulling together on shared things'.* (White British Male, age 76–80, depression symptoms)

Some participants remained depressed or anxious despite having good social links, particularly if they were experiencing more severe symptoms, although many still highlighted the positive aspects of having social support in keeping them going, in contrast to those who were more socially isolated and did not have significant family or friend networks.

*'I mean some of these poor people here [in sheltered housing block], they really are on their own, full stop, yes. And I don't know how they do it.'* (White British Male, age 86–90, depression symptoms)

**Group identity and autonomy.**   Interestingly, attitudes to formalized peer support groups or Day Centers, where people were grouped on the basis of age alone, were much less positive, as they presented a negative stereotype of old age that participants did not want to identify with or be labeled as. Participants often used downward social comparisons as a coping mechanism, comparing themselves favorably to other older adults who had lost their independence, and who seemed to represent 'the other' they feared turning into. Many participants therefore expressed resistance to being labelled as 'old', did not identify with others in the group, or were frequently irritated by them, although it was interesting to note that some who expressed this view had not accessed these groups or had low awareness of what was available.

*'The girls keep saying to me, 'Go to the old people's clubs'. And to be honest with you, I wouldn't want to go up there and sit with all them farts and you know eat fish and chips or a roll and coffee. It's not me'.* (White British Male, age 86–90, depression symptoms)

Some participants displayed a preference for being around younger people, who they felt were less likely to 'make them miserable'. Being able to have autonomy over whom they talked to, where and when, seemed to be an important aspect of 'informal' support that made it work for them, and many disliked the artificiality of organized groups. Some participants felt that setting up more informal or ad-hoc environments or spaces, such as that provided within sheltered housing, might be useful, as these had fewer negative stereotypes attached to them.

*'Yes, I think there should be more places set up where people can go and talk'.* (White British Female, age 76–80, depression symptoms)

Shared interest groups were also mentioned as potentially useful and were more acceptable than 'old people's groups'. Participants tended to rely on recommendations from friends and family, rather than actively seeking out information about local groups themselves.

**Internal mental and emotional coping strategies.** Participants demonstrated a range of internal strategies to deal with their symptoms of depression and anxiety. These strategies were varied and depended on participants' personalities, values and life experiences.

**Drawing on life experiences.** Many participants reported a range of previous traumatic life experiences, including evacuation and bombing during the Blitz, family bereavements (with four participants having experienced the death of adult children), and physical, sexual and emotional abuse.

Many highlighted that their generation's way of dealing with trauma was characterized by carrying on regardless and not discussing or sometimes even acknowledging painful or traumatic events, often described as being 'stoic'. For some, this underpinned a positive way of self-managing depressive symptoms, drawing on strength gained from previous difficult life experiences, resulting in feelings of resilience in the face of future challenges. Because they had survived more severe episodes in the past, they felt a sense of familiarity with depression and knew from experience that periods of low mood were likely to pass with time if they rode it out.

*'I think we've all come from a generation that's fairly stoic, with the war years and that. . .Mum didn't have to do everything for you and all the rest of it. And so when these things come along, you've got that experience, it helps you cope'.* (White British Male, age 86–90, depression symptoms)

However, others felt that this approach was not helpful, particularly within family contexts, and that not dealing with the trauma at the time had caused them to repress painful memories, which consequently had lasting negative effects on their mental health.

*[Referring to her brother's suicide] 'We didn't deal with it as a family. . .at one point I was trying, I was saying that I thought it would be really good for us to have some family therapy. But they wouldn't all agree.'* (White British Female, age 71–75, depression and anxiety symptoms)

Another way of coping, particularly in the face of functional decline or other external factors, was accepting that some things were outside of their control.

*'But there are certain things that you have no power over. And I think when you realize that, you can, you can be more content'.* (White British Male, age 76–80, depression and anxiety symptoms)

In contrast to a more 'stoic' approach, some participants preferred to accept and express their feelings as a positive coping strategy, where being allowed to occasionally wallow or have a 'good cry' were helpful outlets for their feelings that they could then move beyond.

*'Sometimes I think oh what's the matter with me and then I sort of get that feeling–it's a frightened feeling. And then I cry and then I feel better.'* (White British Female, age 81–85, depression and anxiety symptoms)

Drawing on life experiences appeared to be employed less for anxiety symptoms that were exacerbated by newer health concerns, a sense of irreversible physical decline and existential

fears about the future (e.g. constant worrying about preparedness around death and dying, financial cost of funerals, sorting out inheritance)—worries that were much less familiar, and for which they needed to develop new coping strategies.

*'I think, "Ooh, I'd better get the will sorted out.". . ...Silly things that really don't matter because I won't be here anyway. I want it all to run smoothly and [for my children] to get their fair share of what's due to them.'* (White British Male, age 76–80, depression and anxiety symptoms)

*'I know we're getting old and I can see the future is not good ahead. And I just can't see where we're going and what we're doing.'* (White British Female, age 86–90, depression & anxiety)

**Identifying and addressing the cause.** Many of those living with depression felt that they were aware of the causes of their depression, particularly if they had experienced previous episodes. They often demonstrated pragmatic outlooks, locating perceived solutions within their internal control (e.g. rationalizing, taking action) rather than in external interventions (e.g. psychological therapies or antidepressants).

*'If it's, you know, the reason why you're depressed you try to look at the, the cause and you know, focus on that really, and not rely on medication'.* (White British Female, age 76–80, depression symptoms)

*'But I think I can rationalise it a bit more now than I used to. You know, I've had that much more experience. And, of course, that earlier [depressive] episode in life has taught me a lot.'* (White British Male, age 86–90, depression symptoms)

However, those who reported experiencing more severe depression and/or anxiety often expressed a sense of hopelessness or resignation, particularly if they lacked social support, as they were aware of the causes but were unable to resolve these (e.g. past trauma, family conflict). Anxiety, in particular, seemed more difficult to rationalize, and participants had difficulty recognizing, processing and gaining perspective on their symptoms. Often, they could not identify why they felt anxious, which meant they could not address its cause, or it was a newer, ageing-related anxiety (e.g. fear of falling, death, deteriorating health, worries about getting dementia) for which they did not have pre-developed coping strategies for, or they minimized it and put it down to 'silly little things'. Consequently they often downplayed their symptoms and rarely sought help unless their symptoms were severe.

*'Now I find that the slightest thing, it doesn't matter what it is, the smallest thing, which is ridiculous, is enough for me to then start to worry.'* (White British Male, age 76–80, depression & anxiety symptoms)

*'But I am terrified at not dying and people thinking I'm dead. . . .And being buried alive. Or even being in a coffin alive or even being in a body bag alive. And in fact generally I feel it's a claustrophobic thing. That can actually bring on really strong feelings of panic and phobia, which is irritating you know, because it's very irrational.'* (White British Male, age 81–85, depression and anxiety symptoms)

**Distraction and avoidance.** Nearly all participants described using distraction and avoidance to not dwell on things that made them feel sad or anxious; for example, watching

television, playing Mahjong or doing crossword puzzles. Whilst most accepted that this was not a long-term fix, distraction was employed to a greater degree by those who felt that their problems lacked solutions, for example, those with chronic pain, financial problems or multiple health issues.

> '*As soon as I start playing Mahjong, I forget what's going on. I forget my [chronic] pain. I forget I'm down. . .*' (White British Female, age 76–80, depression symptoms)

> '*I just put the television [on] to keep my mind off [financial problems, multiple chronic health conditions]. I just watch television and I, I don't even remember, I put the television so loud.*' (South Asian Female, age 76–80, depression and anxiety symptoms)

Avoidance of triggering situations (such as going out or using public transport) appeared to be viewed as a short-term solution by some, especially those experiencing anxiety, but could lead to increased isolation and was not viewed as helpful in the long-term.

> '*But because that is my fear, you know. I don't, wouldn't like to go out and anything, anything happen, and I'm outside on the road and nobody don't know who I am. And so that's why I stay indoors.*' (Black Caribbean Female, age 76–80, anxiety symptoms)

**Having faith.** Eight female participants, nearly all of whom were from Christian backgrounds, mentioned their faith as important in helping them cope with their depression and /or anxiety symptoms. Some felt being *listened to* by God and being able to have an internal dialogue were useful, whilst others felt that being able to see their lives as part of God's plan and having trust in Him were important in gaining perspectives on their situations, provided them with a sense of meaning and facilitated acceptance of their situation. Interestingly, most participants cited the personal aspect of faith, rather than the social aspect (e.g. going to Church or being part of a congregation), as being the most helpful. This may reflect a combination of previously outlined strategies, such as meaning-making and acceptance. For some, the act of prayer in itself was seen as therapeutic.

> '*By the time I've finished with that prayer, I'm well.*' (Black Caribbean Female, age 76–80, anxiety symptoms)

> '*I think my Christian faith and getting grace of strength that helped me keep going through it all.*' (White British Female, age 76–80, depression symptoms)

## Discussion

We interviewed 28 frail older adults (aged 75+), who described a number of strategies for self-managing depression and/or anxiety symptoms. Our findings provide novel insights into the way in which mood is self-managed in this population, on both a theoretical and practical level, as outlined in more detail below.

Maintaining independence and engaging in activities that promoted a sense of meaning-making were important in nurturing a sense of identity, which participants feared losing as part of their frailty. Meaningful activities for frailer older people were often those that were creative or productive, such as craftwork, photography, music and writing, which could easily be undertaken at home in the context of their frailty. In our study, frail older adults found fostering their own social connections important in promoting a sense of connectivity, which helped

them manage their mood. This was facilitated through local groups and increased use of phones and social media where participants left the home less often. Internal mental and emotional coping strategies employed were varied and shaped according to earlier life experiences, values and contexts. Drawing on life experiences could foster a sense of resilience towards current and future adverse circumstances for some, but for others could lead to unresolved issues. There appeared to be a discrepancy between generational 'stoic' attitudes and how these sometimes translated to individual and family contexts, suggesting that discourses around stoicism might lead to inaccurate understandings of how older people manage mental health symptoms in reality [65]. Attempting to address underlying causes was used to manage depressive symptoms, but anxiety was harder to rationalize and cope with as it rarely had a clear cause. Participants also used distraction and avoidance, which was helpful for severe symptoms and situations that could not be changed, but this could be maladaptive if it stopped people doing activities such as going out. Faith was a significant coping mechanism for a minority of participants.

On a theoretical level, our findings suggest that existing models of ageing do not relate well to the oldest-old; activity theory [66] purports that older adults are most happy when active, failing to take into account that some activities may be limited by frailty in the oldest-old [67], whilst disengagement theory [68] suggests that older adults seek to reduce their social connections and activities as they get older. A recent study [69] presents an alternative 'engage-disengage model' for use in the oldest-old that, whilst recognising that many activities may no longer be achievable for frailer populations, prioritizes engaging in behaviours and activities that can improve well-being and quality of life, and disengaging from those that do not or are no longer achievable. Findings from our study support the use of such a model for frail older adults experiencing mood disorders, as participants adapted their activities in accordance with their frailty, whilst simultaneously also prioritizing maintaining social connections and a sense of independence. Our findings are also in keeping with Baltes et al's (1990) [70] selective optimisation with compensation theory of successful ageing, which states that there are three components of optimal adaptation to the challenges of ageing–selection, optimisation and compensation. Selection involves selecting or prioritising goals in response to a loss in resources or functioning; optimisation refers to optimising goal-related activity by learning new skills or practicing skills; and compensation involves findings ways to compensate for losses in resources or functioning (e.g. by using aids and adaptations or asking others for help).

Our study is also relevant to quality of life models used in older populations, such as the CASP-12 [71], which places emphasis on control, autonomy, self-realisation and pleasure. Our findings suggest these measures of quality of life are applicable to older-old populations experiencing depression and anxiety. In line with a recent study [71], the emphasis on autonomy was especially evident in our findings as being central to maintaining a sense of identity, as well as self-realisation through engaging in meaningful activities, although pleasure was interestingly not highlighted by our participants as a main motivating factor.

On a practical level, our findings suggest that frail older people use similar strategies to self-manage low mood as robust younger-old people, with sociability, self-efficacy and life experience being important themes [25, 53, 54]. Frailty limited the use of some self-management strategies, such as going out or exercising, but participants were able to find alternative activities that were meaningful to them. This is in agreement with existing research that meaningful activities reduce depression in older adults [72] and that meaning is an important enabler for self-management [55]. However, previous studies have focused mainly on outside interventions or ways to increase engagement with existing services. In line with the recent literature [42, 55, 56], results from our study suggest that interventions should facilitate older adults to draw on the wealth of internal mental and emotional coping strategies utilized to manage their

mood without accessing formal support; a finding that has also been highlighted as important in the management of loneliness in older adults [73].

Whilst previous research has suggested that self-management may be helpful for anxiety [74], this relies on older adults having the tools to successfully manage anxiety. A major finding from our study suggests that anxiety, often underpinned by newer health fears and worries about the future, may be more difficult to self-manage than depression as most frail older adults interviewed had not developed effective coping strategies for this other than distraction and avoidance [75]. Only 2 participants in our sample mentioned relaxation techniques such as controlled breathing and meditation, suggesting simple advice or resources to raise awareness of these kind of techniques may be helpful. Findings from a sister paper [61] have also highlighted uncertainty about the validity of seeking professional help for anxiety amongst frail older adults. The conclusion from both papers, in keeping with other international qualitative studies on anxiety in older adults [76, 77], suggests more work needs to be done to raise awareness of anxiety as a diagnosis and find helpful ways of ameliorating anxiety symptoms in this population.

## Strengths and weaknesses

We captured a wide range of views from older adults from a variety of socioeconomic backgrounds, degrees of frailty and symptom severity. We placed an equal focus on both symptoms of anxiety and depression, which are often interconnected and demonstrate high levels of comorbidity [78]. Our topic guide and thematic framework were developed with a range of team members and PPI perspectives, providing a rigorous analysis process, incorporating both clinical and non-clinical, professional and non-professional perspectives. We found no noteworthy differences in self-management according to socioeconomic background or gender. However, we had a low response rate from more severely frail older adults (e.g. from nursing homes), older men, ethnic minority groups and non-English speakers. In our sample, relatively few reported a formal diagnosis of anxiety and/or depression, which may reflect low rates of help-seeking and diagnosis in this population, or that the experiences of those with more severe symptoms may not have been fully explored. We were unable to access participants' medical notes to verify self-reported diagnoses, and we did not collect specific data on current or previous treatments, although about two thirds of our sample reported having tried antidepressants. Furthermore, as participants were all currently experiencing some degree of low mood and/or anxiety symptoms, the longer-term impact of these self-management strategies was unclear, and we did not objectively assess symptom severity. It is therefore possible that individuals may perceive their strategies to be effective when they are not in the longer-term, and future prospective studies are needed to explore the effectiveness of different self-management strategies. It is also important to note that the results of this study reflect the views of a high-income Western country with free national healthcare and may not be transferrable to lower-income countries, which may have higher levels of frailty and more limited resources [79]. Furthermore, data were collected prior to the Covid-19 pandemic, and may not accurately reflect the additional impact on mental health from lockdowns or shielding, concerns around the risk of contracting Covid-19 and limitations on group activities (for example, many social groups ceased activities or moved online during the pandemic). It may also not reflect the full range of preferred self-management strategies of frail older adults today (for example, a potentially greater use of technology amongst the older adult population [80]).

## Implications

Information and advice for frail older adults to address depression and anxiety should communicate techniques that are acceptable and helpful (e.g. maintaining routines, meaningful activities, finding ways to socialize despite limited mobility), as well as highlighting the benefits

and limitations of self-management techniques that may vary according to personality, situation and symptoms (e.g. acceptance, stoicism, distraction). Frail older adults should be encouraged to reflect on what has helped them in the past and what is meaningful for them. Further awareness needs to be raised regarding anxiety and ways in which frail older people can self-manage such symptoms. However, as frail older adults rarely seek mental health information from sources other than their GP [27, 61] (whom they may delay seeing until symptoms are severe), multiple avenues are likely to be needed to communicate these, such as through voluntary community and social enterprise (VCSE) organizations, healthcare settings (e.g. waiting room posters/leaflets, pharmacists) and interactions with healthcare professionals.

GPs might help frail older adults maintain their independence for longer by early referral to multidisciplinary services and relevant social prescribing services. As in other studies, we found that meaningful social activities were important [81], and that older adults preferred informal social groups of their own choosing rather than formalized groups which collectively labelled them as 'old' [82, 83]. Previous studies have highlighted a high degree of internalized ageism in older adults' attitudes [25], which is often associated with negative health outcomes [84]. Our study highlighted that autonomy and choice over who they talked to and in what environment was important for our participants' sense of group identity. Creating community spaces for frail older adults to get together or offering inter-generational groups that center around a common interest may be useful for improving wellbeing in this group. For those with mobility limitations, these spaces may need to be incorporated closer to home, e.g. within sheltered housing, with encouragement to use remote forms of communication such as phone and video calling. VCSE workers would be well-placed to help frail older adults identify available appropriate services and could receive training to provide basic advice on self-management techniques. A recent study looked at the benefits of a guided self-management approach for anxiety in older veterans [85]. This may help with recognition and understanding of these symptoms which were clearly troubling for many of our participants.

## Conclusion

There is an urgent need for mental health support for older adults, given the context of an increasingly ageing global population, and the need is even greater in wake of the Covid-19 pandemic [20], which has also put additional pressures on mental health services and increased waiting times [23]. For frail older adults with mood disorders, self-management may have an important and acceptable role in maintaining wellbeing, preventing relapse and providing an adjunct to other treatments, as well as offering an alternative to formal treatment for those with less severe symptoms, those who are on waiting lists for therapy or who are not able or willing to engage with services. Our research suggests that frailty limits some strategies used to self-manage mood, but that people adapt their self-management techniques to account for this, for example, focusing more on sedentary, creative pursuits or maintaining routines. Helping frail older adults maintain independence and engage in meaningful and social activities were all identified as potentially important in the self-management of depression and anxiety in this population. Supporting frail older adults' internal coping strategies may be useful for dealing with depression; however, those with anxiety may need further advice and techniques to manage their symptoms. Self-management support is likely to be more helpful when tailored to the individual, taking into account personal preferences, life experiences, levels of functioning and available resources.

## Acknowledgments

We would like to thank our PPI representatives, Wes Dowridge and Shamime Lakda, for their valuable input throughout this project.

## Author Contributions

**Conceptualization:** Pushpa Nair, Kate Walters, Rebecca Gould, Kalpa Kharicha, Marta College Buszewicz, Rachael Frost.

**Formal analysis:** Pushpa Nair, Kate Walters, Su Aw, Rebecca Gould, Kalpa Kharicha, Marta College Buszewicz, Rachael Frost.

**Funding acquisition:** Pushpa Nair, Kate Walters, Rebecca Gould, Kalpa Kharicha, Marta College Buszewicz, Rachael Frost.

**Investigation:** Pushpa Nair, Rachael Frost.

**Project administration:** Pushpa Nair, Kate Walters, Marta College Buszewicz, Rachael Frost.

**Supervision:** Kate Walters, Marta College Buszewicz, Rachael Frost.

**Validation:** Pushpa Nair.

**Visualization:** Pushpa Nair, Rachael Frost.

**Writing – original draft:** Pushpa Nair.

**Writing – review & editing:** Pushpa Nair, Kate Walters, Su Aw, Rebecca Gould, Kalpa Kharicha, Marta College Buszewicz, Rachael Frost.

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
