## [Decision Letter · Decision Letter 0]

25 Aug 2022

PONE-D-22-04184Self-management of depression and anxiety amongst frail older adults in the United Kingdom: a qualitative studyPLOS ONE

Dear Dr. Nair,

Thank you for submitting your manuscript to PLOS ONE. After careful consideration, we feel that it has merit but does not fully meet PLOS ONE’s publication criteria as it currently stands. Therefore, we invite you to submit a revised version of the manuscript that addresses the points raised during the review process.

Please refer to the second reviewer's comments in particular.  I felt that these were thoughtful and constructive and required some input==============================

We look forward to receiving your revised manuscript.

Kind regards,

Adrian Stuart Wagg, MD

Academic Editor

PLOS ONE

https://journals.plos.org/plosone/s/file?id=ba62/PLOSOne_formatting_sample_title_authors_affiliations.pdf".

2. We noted in your submission details that a portion of your manuscript may have been presented or published elsewhere. [Frost R, Nair P, Aw S, Gould RL, Kharicha K, Buszewicz M, Walters K. Supporting frail older people with depression and anxiety: a qualitative study. Aging Ment Health. 2020 Dec;24(12):1977-1984. doi: 10.1080/13607863.2019.1647132] Please clarify whether this [conference proceeding or publication] was peer-reviewed and formally published. If this work was previously peer-reviewed and published, in the cover letter please provide the reason that this work does not constitute dual publication and should be included in the current manuscript.

Reviewers' comments:

Reviewer's Responses to Questions

**Comments to the Author**

1. Is the manuscript technically sound, and do the data support the conclusions?

Reviewer #1: Yes

Reviewer #2: Partly

2. Has the statistical analysis been performed appropriately and rigorously? 

Reviewer #1: N/A

Reviewer #2: N/A

3. Have the authors made all data underlying the findings in their manuscript fully available?

Reviewer #1: Yes

Reviewer #2: No

4. Is the manuscript presented in an intelligible fashion and written in standard English?

Reviewer #1: Yes

Reviewer #2: Yes

5. Review Comments to the Author

Reviewer #1: Thank you for giving me the opportunity to review what is an excellent paper about a critical area of health care. It is well written and cogently argued. Abstract is representative of paper's content. Introduction is thorough, with an excellent literature review. Methodology is appropriate for research questions. Data analysis is thorough and representative of participant voices. Discussion draws comparisons between results and previous research. Conclusions are justifiable based on data. Limitations are appropriately identified, and implications for practice and future research are clearly articulated.

Very small number of missing references and a couple of possible grammatical errors.

Reviewer #2: Dear authors,

I read your manuscript with great interest. Overall, this is a well written piece of research that focuses on the mental health needs of a chronically underserved group. I felt the background sections was excellent and justified the research well. Methods were generally sound.

However, I felt the findings section needs further consideration to help highlight originality. There was some overlap between themes and the data seems "thin" in places, which means important context may be lacking. The discussion could also be strengthened with further critique of a wider selection of evidence and, crucially, consideration of some relevant theory.

I provide the following points to help support revisions, I hope you find these helpful and constructive. Please do not be disheartened by the number of suggestions, this manuscript has great potential. I just felt more could be done to tease out the novelty.

Background

Ln 118: Can the authors be more specific about what “going out” means. Seems a bit colloquial and would be assumed to overlap with socializing etc.

Research design and methods

Ln135-138 is a little “clunky”. Consider revising and avoid homogenising older adults, which is implied when using language such as “they" and "them” (check remaining manuscript for instances of this).

Data collected in 2018 – so all pre-covid. This should be discussed in the limitations.

There were quite a few members of the team involved in the analysis. It would be interesting to reflect on this in strengths and limitations.

The two PPIE members are described as representing diverse backgrounds. Given there are only two members, describing these as from ‘different’ backgrounds may be more appropriate.

Saturation at 26 interviews seems quite a high number based on the depth of the current analysis described in the paper. I understand that other themes are reported in another paper. Is this paper a re-analysis or were all themes identified in a single process of analysis? Were any themes not reported?

Results

Impact of frailty - this theme sits a part to all others, which describe self-management strategies. Is this more of a central organising theme/concept? I think it would add further clarity to describe it in these terms, as frailty provides the context to self-management strategies. The self-management strategies also suggest some higher order concepts and themes.

Self-management strategies:

Overall, I found the self-management strategy theme labels to be overly descriptive. Many miss a reference point to help understand the context e.g. Control (of/over what?), acceptance (of what?).

Maintaining a routine – interpretation under this sub-theme refers to downward social comparisons, but this lacks supporting data and feels more relevant to internal/external coping strategies. This sub-theme also refers to maintaining independence, which would seem to be the higher order concept at play here. Maintaining a routine could be a means by which participants demonstrate their independence, despite experiencing frailty.

Meaningful and creative activities - the data refer to craft (which evokes memories of the past), history projects, genealogy. Do these activities have a point of connection? In what ways are they meaningful? Tend to see theme labels like “engaging in meaningful routines and activities’ quite a lot. Again, the higher-order concept here is meaning making, in which creative activities are a means to achieve meaning.

Ln365: Not sure I understand the reference to altruism. The data suggests a sense of togetherness, belonging, shared group identity rather than altruism/charity.

Ln 386-389: The data here seem to relate to identity, being defined as “old” and a stereotypical view of “old people’s clubs" – suggests a lack of awareness about support services and opportunities. How does this relate to control? There is overlap here with independence as signalled in an earlier theme (and likewise autonomy, which is not discussed).

Ln 395: How does the corresponding data indicate “relaxed” environments are preferred? And what might “relaxed” mean?

Ln 401: The authors describe shared interest groups. Again, how does this relate to control? This does not seem to be the main or indeed only issue/concept at play here. Control might fit better as an internal coping strategy - you will have a better understanding of the data than I.

What did participants understand/mean by the term stoicism – could the authors tease this out more, what did this mean to individuals? This term needs to be approached critically along with resilience, as both are over-used particularly within research on this cohort.

The two contrasting quotes are interesting (Ln 420-423 and Ln 429-432) in that the first talks at a general level about the generation (We with a capital W, I suppose), whereas the second refers to a family context (we with a lowercase w) and how this specific tragedy challenged the broader view of the stoic generation. It would be helpful to consider this in the discussion.

Ln 434 – 447: To what extent does this data relate to symptoms of anxiety? Are these worries about mortality and preparedness for death actually symptoms of anxiety? Or did participants demonstrate constant worry about these topics?

Is there enough data here for a sub-theme on acceptance? The data seem to recognise a potential end-point for stoicism/resilience/control (perhaps an "emptying" of the resilience reserves)- perhaps this data could be amalgamated under other headings?

Ln 482: What do the authors mean by “realistic anxiety”? An interesting term. Suggests other types of anxiety are unrealistic and therefore not real.

Could the theme on having faith be expanded? How might faith relate to other themes and sub-themes? Being part of a faith group and/or congregation may be an important source of meaning.

Discussion

The findings seem to corroborate a lot of what we already know. Could the authors do more to signal the novelty and originality of this manuscript in relation to the existing evidence base?

A broader consideration of the literature would help to add important critique to the discussion of findings. There is also a lack of theoretical development and consideration. The findings potentially talk to quality of life models that were developed for older adults e.g. CASP (Control, Autonomy, Self-Realisation, Pleasure). Other gerontological literature may also be relevant here in terms of activity and adaptation theories in later life (e.g. disengagement, compromise) - the context of frailty and mental health could add new contexts and insights here. Consideration of some theory would be good here.

6. PLOS authors have the option to publish the peer review history of their article (what does this mean?). If published, this will include your full peer review and any attached files.

Reviewer #1: No

Reviewer #2: No

---

## [Author Response · Author response to Decision Letter 0]

24 Oct 2022

*******PLEASE SEE ATTACHED FILE 'RESPONSE TO REVIEWERS******** (below is a copy and paste from that file)

REVIEWER 1

• Reviewer #1: Thank you for giving me the opportunity to review what is an excellent paper about a critical area of health care. It is well written and cogently argued. Abstract is representative of paper's content. Introduction is thorough, with an excellent literature review. Methodology is appropriate for research questions. Data analysis is thorough and representative of participant voices. Discussion draws comparisons between results and previous research. Conclusions are justifiable based on data. Limitations are appropriately identified, and implications for practice and future research are clearly articulated.

Very small number of missing references and a couple of possible grammatical errors.

Thank you for your comments. We are pleased that you found our paper interesting and valuable. We have double checked the references and grammar, as suggested.

REVIEWER 2

• Reviewer #2: Dear authors,

I read your manuscript with great interest. Overall, this is a well written piece of research that focuses on the mental health needs of a chronically underserved group. I felt the background sections was excellent and justified the research well. Methods were generally sound.

However, I felt the findings section needs further consideration to help highlight originality. There was some overlap between themes and the data seems "thin" in places, which means important context may be lacking. The discussion could also be strengthened with further critique of a wider selection of evidence and, crucially, consideration of some relevant theory.

I provide the following points to help support revisions, I hope you find these helpful and constructive. Please do not be disheartened by the number of suggestions, this manuscript has great potential. I just felt more could be done to tease out the novelty.

Thank you for your very helpful and insightful comments. We have addressed them to the best of our ability, as outlined point-by-point below. (Line numbers relate to the tracked changes version of the manuscript).

• Background

Ln 118: Can the authors be more specific about what “going out” means. Seems a bit colloquial and would be assumed to overlap with socializing etc.

Thank you for highlighting this. As you rightly point out, there is an overlap with socializing. However, by ‘going out’, we also more broadly meant activities that promoted people leaving the house, e.g. going shopping. We have therefore changed the wording to more accurately reflect this (Ln 123-4).

• Research design and methods

Ln135-138 is a little “clunky”. Consider revising and avoid homogenising older adults, which is implied when using language such as “they" and "them” (check remaining manuscript for instances of this).

Thank you for highlighting this. We have reworded this section to avoid using homogenising language and have also checked the rest of the manuscript for any other instances of this (Ln 141-44).

• Data collected in 2018 – so all pre-covid. This should be discussed in the limitations.

Thank you for highlighting this important point. This point, and reflections related to it, have now been included in the ‘Strengths and Weaknesses’ section (Ln 768-74).

• There were quite a few members of the team involved in the analysis. It would be interesting to reflect on this in strengths and limitations.

Thank you for highlighting this. Reflections related to this point have now been included in the ‘Strengths and Weaknesses’ section (Ln 744-46). 

• The two PPIE members are described as representing diverse backgrounds. Given there are only two members, describing these as from ‘different’ backgrounds may be more appropriate.

Thank you, we have amended the wording as you suggested (Ln 198-99).

• Saturation at 26 interviews seems quite a high number based on the depth of the current analysis described in the paper. I understand that other themes are reported in another paper. Is this paper a re-analysis or were all themes identified in a single process of analysis? Were any themes not reported?

We found that data from the interviews was very rich and whilst we were doing the analysis, we realised that two distinct sets of themes were emerging from the data, each of which were worthy of discussion in a paper in their own right. A sister paper therefore reports on themes relating to attitudes /experiences of seeking professional support and treatments (such as antidepressants and psychological therapies), which was the initial focus of this study. However, because so many participants reported using self-management strategies, we decided to report themes related to this in a second paper (this current paper being reviewed). Data were coded in a single process, line-by-line, identifying all emergent themes in an overarching thematic framework. The thematic analysis (led by RF) focusing on treatments delivered by professionals, was carried out initially and findings reported in the first paper (Frost, Nair, Aw, Gould, Kharicha, Buszewicz & Walters (2020), Supporting frail older people with depression and anxiety: a qualitative study, Aging & Mental Health, 24:12, 1977-1984). A subsequent thematic analysis was conducted for this second paper (led by PN), focusing on self-management. All major themes that emerged from the data were reported across the two papers. We have clarified the relationship between the 2 studies on page 8 (Ln 204-07).

• Results

Impact of frailty - this theme sits a part to all others, which describe self-management strategies. Is this more of a central organising theme/concept? I think it would add further clarity to describe it in these terms, as frailty provides the context to self-management strategies. The self-management strategies also suggest some higher order concepts and themes.

Thank you for your insightful comment. We have decided to remove ‘impact of frailty’ as a theme, and instead merge it within the Results section, as it sets the context for the self-management strategies described.

• Self-management strategies:

Overall, I found the self-management strategy theme labels to be overly descriptive. Many miss a reference point to help understand the context e.g. Control (of/over what?), acceptance (of what?).

Thank you for highlighting this. We have renamed the themes and subthemes, so that they more accurately depict and help the reader understand the context being described (see Results section). The 4 main themes are now: 1. Maintaining independence; 2. Meaning-making and recreation; 3. Socializing and peer support (subthemes: camaraderie and connectivity; group identity and autonomy); 4. Internal mental and emotional coping strategies (sub themes: drawing on life experience; identifying and addressing the cause; distraction and avoidance; faith).

• Maintaining a routine – interpretation under this sub-theme refers to downward social comparisons, but this lacks supporting data and feels more relevant to internal/external coping strategies. 

Thank you for your comment. On reviewing this point, we agree that it does not fit well in this section. We have therefore instead integrated it into the ‘Socializing and peer support’ theme, under the subtheme ‘Group identity and autonomy’, whereby participants used downward social comparisons as a coping mechanism, preferring to look down on other ‘stereotypical’ frail older people, rather than identify with them (Ln 436-45).

• This sub-theme also refers to maintaining independence, which would seem to be the higher order concept at play here. Maintaining a routine could be a means by which participants demonstrate their independence, despite experiencing frailty.

Thank you for your helpful comment. We agree that maintaining independence is the higher order theme here and have therefore changed the name and focus of this theme accordingly to reflect this (Ln 258-341).

• Meaningful and creative activities - the data refer to craft (which evokes memories of the past), history projects, genealogy. Do these activities have a point of connection? In what ways are they meaningful? Tend to see theme labels like “engaging in meaningful routines and activities’ quite a lot. Again, the higher-order concept here is meaning making, in which creative activities are a means to achieve meaning.

Thank you for highlighting this. On reviewing this theme, we feel that ‘maintaining independence’ and ‘meaning-making and recreation’ are two separate themes, and have therefore split them into two (pages 11 and 12). Within meaning-making, as you point out, creative activities are a means to achieve meaning. We have also broadened this section to include recreation in general, in recognition that some participants may engage in some of these activities purely for leisure/to pass the time, rather than trying to ‘make sense’ of things (Ln 343-71).

• Ln365: Not sure I understand the reference to altruism. The data suggests a sense of togetherness, belonging, shared group identity rather than altruism/charity.

We have removed the term ‘altruism’ and replaced with ‘a sense of shared experiences’ (Ln 418-19).

• Ln 386-389: The data here seem to relate to identity, being defined as “old” and a stereotypical view of “old people’s clubs" – suggests a lack of awareness about support services and opportunities. How does this relate to control? There is overlap here with independence as signalled in an earlier theme (and likewise autonomy, which is not discussed).

Thank you for highlighting this. On reviewing this section, we agree that control is not the main driver here. We have therefore renamed this theme ‘group identity and autonomy’, and sought to highlight the importance of identity, and its close link to independence and autonomy for this group. We have also highlighted the lack of awareness of local service provision and the fact that some participants expressing negative views about ‘old people’s clubs’ had not accessed these (Ln 435-62).

• Ln 395: How does the corresponding data indicate “relaxed” environments are preferred? And what might “relaxed” mean?

• We have changed the wording here to ‘informal or ad-hoc’ – settings that feel more natural, with fewer attached pre-conceived notions or stereotypes associated with age (Ln 460-62).

• Ln 401: The authors describe shared interest groups. Again, how does this relate to control? This does not seem to be the main or indeed only issue/concept at play here. Control might fit better as an internal coping strategy - you will have a better understanding of the data than I.

Thanks again for highlighting this. As mentioned above, we agree that control is not the main concept at play here and have removed it from the theme accordingly. Rather, we feel that group identity and autonomy are the main concepts here, and have renamed this section to reflect this (Ln 435-69).

• What did participants understand/mean by the term stoicism – could the authors tease this out more, what did this mean to individuals? This term needs to be approached critically along with resilience, as both are over-used particularly within research on this cohort.

Thank you for highlighting this. We have further elaborated on how participants described their generation’s approach to coping, for which some participants explicitly used the term ‘stoicism’ and others did not, and how this impacted on their subsequent coping styles (Ln 483-504). We have also clarified what we mean by resilience. However, we have taken both these terms (‘stoicism’ and ‘resilience’) out of the theme heading and replaced with ‘Drawing on life experiences’ (Ln 475), in recognition that these terms are overused, and to enable us to provide more context when using them within the text of the subtheme.

• The two contrasting quotes are interesting (Ln 420-423 and Ln 429-432) in that the first talks at a general level about the generation (We with a capital W, I suppose), whereas the second refers to a family context (we with a lowercase w) and how this specific tragedy challenged the broader view of the stoic generation. It would be helpful to consider this in the discussion.

Thank you for raising this very interesting point. We have considered this in the discussion (Ln 672-75).

• Ln 434 – 447: To what extent does this data relate to symptoms of anxiety? Are these worries about mortality and preparedness for death actually symptoms of anxiety? Or did participants demonstrate constant worry about these topics?

We have reworded this sentence (Ln 533) to make it clear that the worries mentioned here were cited by participants as causing them anxiety (i.e. constant levels of worry/preoccupation).

• Is there enough data here for a sub-theme on acceptance? The data seem to recognise a potential end-point for stoicism/resilience/control (perhaps an "emptying" of the resilience reserves)- perhaps this data could be amalgamated under other headings?

Thank you for highlighting this. We agree that the ‘acceptance’ sub-theme could be incorporated within the other sub-themes in this section, and have amended this section accordingly (Ln 506-30).

• Ln 482: What do the authors mean by “realistic anxiety”? An interesting term. Suggests other types of anxiety are unrealistic and therefore not real.

Thank you for highlighting this. On reflection, we agree that this term is not appropriate, and have reworded this sentence to more accurately reflect newer, ageing-related fears (Ln 588).

• Could the theme on having faith be expanded? How might faith relate to other themes and sub-themes? Being part of a faith group and/or congregation may be an important source of meaning.

Thank you for highlighting this. Whilst we have kept this subtheme under ’internal mental and emotional coping strategies’, we have added to this section in order to link it to some of the previous theme/subthemes and also point out that, for most participants, faith on a personal level was viewed as more helpful than the social aspect of belonging to a congregation or going to Church. These faith-based coping approaches link strongly to previous approaches described such as meaning-making and acceptance (Ln 633-37).

Discussion

The findings seem to corroborate a lot of what we already know. Could the authors do more to signal the novelty and originality of this manuscript in relation to the existing evidence base? A broader consideration of the literature would help to add important critique to the discussion of findings. There is also a lack of theoretical development and consideration. The findings potentially talk to quality of life models that were developed for older adults e.g. CASP (Control, Autonomy, Self-Realisation, Pleasure). Other gerontological literature may also be relevant here in terms of activity and adaptation theories in later life (e.g. disengagement, compromise) - the context of frailty and mental health could add new contexts and insights here. Consideration of some theory would be good here.

Thank you for your comments and useful suggestions of relevant literature and theory. We have now included a discussion of our findings in relation to some of the theoretical models of ageing as you suggest (Ln 681-702), as well as linked them to quality-of-life models used in older adults (Ln 703-710). We have also highlighted throughout the discussion section in what way our findings are novel, on both a theoretical and practical level.

COMMENTS TO EDITOR

Thank you, we have amended the manuscript accordingly.

2. We noted in your submission details that a portion of your manuscript may have been presented or published elsewhere. [Frost R, Nair P, Aw S, Gould RL, Kharicha K, Buszewicz M, Walters K. Supporting frail older people with depression and anxiety: a qualitative study. Aging Ment Health. 2020 Dec;24(12):1977-1984. doi: 10.1080/13607863.2019.1647132] Please clarify whether this [conference proceeding or publication] was peer-reviewed and formally published. If this work was previously peer-reviewed and published, in the cover letter please provide the reason that this work does not constitute dual publication and should be included in the current manuscript.

We have added a statement in the Cover Letter to confirm that this work does not constitute a dual publication, as it reports novel and new findings, which are different and separate from those published in the sister paper.

3. In your Data Availability statement, you have not specified where the minimal data set underlying the results described in your manuscript can be found. PLOS defines a study's minimal data set as the underlying data used to reach the conclusions drawn in the manuscript and any additional data required to replicate the reported study findings in their entirety. All PLOS journals require that the minimal data set be made fully available. For more information about our data policy, please see http://journals.plos.org/plosone/s/data-availability. Upon re-submitting your revised manuscript, please upload your study’s minimal underlying data set as either Supporting Information files or to a stable, public repository and include the relevant URLs, DOIs, or accession numbers within your revised cover letter. For a list of acceptable repositories, please see http://journals.plos.org/plosone/s/data-availability#loc-recommended-repositories. Any potentially identifying patient information must be fully anonymized.

We note that you have indicated that data from this study are available upon request. PLOS only allows data to be available upon request if there are legal or ethical restrictions on sharing data publicly. For more information on unacceptable data access restrictions, please see http://journals.plos.org/plosone/s/data-availability#loc-unacceptable-data-access-restrictions.

We did not seek and do not have approval from NHS REC to disclose this data. Our information provided to participants and consent form state that we will only publish short excerpts from interviews (quotes) and so do not allow for publication of anonymised transcripts. Some of the data provided by participants relates to sensitive issues, such as traumatic experiences (e.g. sexual abuse) and mental health and may not have been disclosed if participants had thought it would be made publicly available. 

We did not seek and do not have approval from NHS REC to disclose these data. Our information provided to participants and consent form state that we will only publish short excerpts from interviews (quotes) and so do not allow for publication of anonymised transcripts. Some of the data provided by participants relates to sensitive issues such as traumatic experiences (e.g. sexual abuse) and mental health and may not have been disclosed if participants had thought it would be made publicly available. As such, we do not believe it is ethical to access the data, which was supplied on the understanding of full confidentiality. Details for the ethics committee for this project are: NHS Camden and Kings Cross Research Ethics Committee (ref 17/LO/1963), camdenandkingscross.rec@hra.nhs.uk

---

## [Decision Letter · Decision Letter 1]

21 Nov 2022

Self-management of depression and anxiety amongst frail older adults in the United Kingdom: a qualitative study

PONE-D-22-04184R1

Dear Dr.  Nair,

We’re pleased to inform you that your manuscript has been judged scientifically suitable for publication and will be formally accepted for publication once it meets all outstanding technical requirements.

Kind regards,

Esmat Mehrabi

Academic Editor

PLOS ONE

---

## [Editor Report · Acceptance letter]

28 Nov 2022

PONE-D-22-04184R1 

Self-management of depression and anxiety amongst frail older adults in the United Kingdom: a qualitative study 

Dear Dr. Nair:

I'm pleased to inform you that your manuscript has been deemed suitable for publication in PLOS ONE. Congratulations! Your manuscript is now with our production department. 

Kind regards, 

on behalf of

Dr. Esmat Mehrabi 

Academic Editor

PLOS ONE